# Probing the Suitability of Different Ca^2+^ Parameters for Long Simulations of Diisopropyl Fluorophosphatase

**DOI:** 10.3390/molecules26195839

**Published:** 2021-09-26

**Authors:** Alexander Zlobin, Igor Diankin, Sergey Pushkarev, Andrey Golovin

**Affiliations:** 1Faculty of Bioengineering, Lomonosov Moscow State University, 119234 Moscow, Russia; diankin@fbb.msu.ru (I.D.); spush@fbb.msu.ru (S.P.); 2Shemyakin and Ovchinnikov Institute of Bioorganic Chemistry, Russian Academy of Sciences, 117997 Moscow, Russia; 3Sirius University of Science and Technology, 354340 Sochi, Russia

**Keywords:** molecular dynamics, metadynamics, QM/MM, DFPase, force field, substrate binding, diisopropyl fluorophosphatase

## Abstract

Organophosphate hydrolases are promising as potential biotherapeutic agents to treat poisoning with pesticides or nerve gases. However, these enzymes often need to be further engineered in order to become useful in practice. One example of such enhancement is the alteration of enantioselectivity of diisopropyl fluorophosphatase (DFPase). Molecular modeling techniques offer a unique opportunity to address this task rationally by providing a physical description of the substrate-binding process. However, DFPase is a metalloenzyme, and correct modeling of metal cations is a challenging task generally coming with a tradeoff between simulation speed and accuracy. Here, we probe several molecular mechanical parameter combinations for their ability to empower long simulations needed to achieve a quantitative description of substrate binding. We demonstrate that a combination of the Amber19sb force field with the recently developed 12-6 Ca^2+^ models allows us to both correctly model DFPase and obtain new insights into the DFP binding process.

## 1. Introduction

Diisopropyl fluorophosphatase (DFPase) is an enzyme from the proteome of *Loligo vulgaris* (European squid) [1]. It attracts ever-growing attention from the scientific community across the globe due to its ability to hydrolyze organophosphorus compounds including G-type nerve agents [2,3]. A number of works have been dedicated to both experimental and computational investigation of this activity [4,5]. From the structural perspective, DFPase is a six-bladed propeller harboring two Ca^2+^ cations [6,7]. Both are required for enzyme function; however, only one is directly involved in the positioning of the substrate and catalytic residues while the other is thought to play a structural role [8]. From the computational perspective, efforts were mostly focused on deciphering the exact reaction mechanism by means of hybrid quantum-mechanical/molecular-mechanical (QM/MM) modeling [9,10,11]. Most recent findings provide evidence for a general-base concerted mechanism with attacking water molecule activation by catalytic residue D229 (Figure 1A); however, an opposing notion of a direct nucleophilic attack of D229 oxygen on phosphorus is still sound (Figure 1B). Together with E21 and two asparagines, N120 and N175, D229 coordinates the catalytic calcium ion. The second cation is coordinated by residues D232, L273, and N274 (Figure 1C).

Pronounced organophosphatase activity of DFPase makes it a promising candidate for developing it into a biotherapeutic to treat organophosphate poisoning by means of enzyme engineering [12]. DFPase readily interacts with G-type nerve agents including sarin, cyclosarin, and soman. However, these compounds include asymmetric phosphorus and thus exist in pairs of forms with substantially different levels of toxicity. While S-enantiomers are more poisonous, DFPase shows a pronounced preference for less toxic R forms. Reversal of enantioselectivity, therefore, is a particularly important milestone en route to developing a potent DFPase-based biotherapeutic. Such work was undertaken previously by rationally designing DFPase variants harboring four substitutions [13]. These variants demonstrated reversed enantioselectivity for sarin and cyclosarin in the experiment. However, the initial design idea was derived by considering the reaction mechanism to rely on a direct nucleophilic attack of D229 on phosphorus, and by manually optimizing the productive binding pose for it. It is not obvious though how these sets of substitutions affect the reaction with enantiomers if the general-base mechanism is considered. The absence of such information limits the ability of this work to inform further developments in producing efficient DFPase-based biotherapeutic. The productive binding pose for DFPase was shown to differ as two rivaling mechanisms are considered alongside [11]. The same may be true for G-type agents. Therefore, to carefully investigate the origins of the reversal it is necessary to describe the whole binding process of chiral substrates as well as their conformational landscapes in bound form. 

Investigation of the binding process in detail can be powered by advances in computational techniques built upon enhancing the sampling capabilities of classical molecular dynamics [14,15,16,17,18,19,20]. A number of such approaches are steadily growing together with the overall rise of the availability of computational resources. However, depending on the system, enhanced sampling on the microsecond scale is still required to acquire meaningful qualitative and quantitative insights. Ensuring maximally efficient utilization of computational resources, therefore, is crucial to turn binding/unbinding simulations into a manageable task.

DFPase is an especially challenging enzyme for molecular modeling since it harbors two calcium ions. A correct description of metal cations coordination in principle requires QM treatment [21]. However, for most QM methods in use, even nanosecond-scale simulations are still unreachable. On the other hand, classical MM description of biomolecules commonly allows for 2 fs integration steps and, in some special schemes, even up to 4–7 fs integration steps, making microsecond-scale simulations routine [22,23]. However, every MM model of metal cations is a tradeoff between such speed and desired accuracy. They are most commonly represented as one particle interacting with others via point electrostatics and 12-6 LJ potential, so no explicit specification of particular coordination geometry and numbers exists. The development of novel models for metal cations in the framework of MM force fields is an ongoing endeavor [21,24,25,26,27,28].

Earlier force field-independent multisite dummy models for divalent cations were proposed [24]. Such models are explicitly tuned to simulate desired coordination geometries. They demonstrated good results in several applications, including DFPase modeling [11,29]. However, this comes with a significant drawback of limiting the integration step to a maximum of 1 fs. Taking into account the stated interest in the substrate-binding process of DFPase in dynamics, the possibility to use 2 fs and higher integration step sizes would provide substantial benefit. Thus, in this work, we decided to compare the dynamic behavior of the dummy calcium model with a couple of simpler ones in conjunction with a number of different freely available modern force fields. We show that for DFPase the use of a multisite dummy model not only is redundant but also provides erroneous results. On the other hand, the most accurate model also allows for 2 fs integration steps thus paving the way to long molecular simulations including such of substrate binding. We conclude our work by providing the results of the first-ever demonstration of the DFPase-DFP binding-unbinding process in dynamics.

## 2. Results and Discussion

### 2.1. Equilibrium Simulations of DFPase under Different Treatment

DFPase harbors two Ca^2+^ cations within its structure. Their positions and architecture of corresponding coordination shells are identical throughout all currently known crystallographic models (Appendix A). We speculate that their dynamical stability can be safely implied as well. What is more, the architecture of the catalytic Ca^2+^ site of DFPase is very close to that of paraoxonase 1, implying a high degree of evolutionary optimization for such a motif to serve as a structural and functional unit within the enzyme. MM treatment therefore can be deemed reliable if it captures the dynamical stability of Ca^2+^-binding sites in the course of an MD simulation. A reliable MM setup is a prerequisite for any molecular modeling work aimed at the exploration of conformational space. An example of such is a substrate binding process. It may feature previously unknown states or be realized through induced fit and thus requires a setup striving to achieve both physical correctness and a high degree of agnosticism, that is, freedom from researcher bias.

For our tests, we selected three modern freely available protein force fields, namely Amber19sb, CHARMM36m, and OPLS-AA/M. They were supplemented with five different calcium cation models: one supplied with the force field distribution (further referred to as DEF), a multisite dummy model (DUM), and three 12-6 LJ models for tip3p-FB water from recent work from the Merz group (COM, HFE, and IOD. Refer to Materials and Methods for further details) [25]. This way, we ended up with 15 parameter sets. For each parameter set under consideration, we derive its quality measure as RMSD of the cation and protein atoms that coordinate it (Figure 1C) over the last 10 ns of five independent 50 ns simulations.

We found that most parameter combinations tested fail to correctly describe the architecture of the catalytic Ca^2+^ site (Figure 1D). While some deviation from zero RMSD is to be expected due to thermal fluctuations, by analyzing individual trajectories we discovered that RMSD over 0.5 Å clearly indicates some major disturbance (Appendix A). The only parameter set reproducibly keeping RMSD under this threshold is the combination of the Amber19sb force field with the COM Ca^2+^ model. It clearly outperforms all other tested parameter combinations in correctly capturing the architecture of the catalytic Ca^2+^ site. 

Unsurprisingly, all pre-packaged calcium cation models fail to do so for each force field tested, with the worst result coming from simulations with the OPLS-AA/M force field. These results however can be significantly improved by substituting this metal model with COM or IOD. No notable difference whatsoever was observed by varying the metal model for the CHARMM36m force field. This finding hints at the dominant role of the force field itself in causing the disturbance. 

Alarming results were obtained regarding the modeling of the structural Ca^2+^ site (Figure 1E). None of the 15 combinations tested were able to keep the crystallographic organization in dynamics. On the contrary, MD simulations commonly culminated in a completely different state in terms of the location and interactions of the structural Ca^2+^ (Figure 2 and Appendix A). However, high dynamism or alternating coordination shells of this cation are not to be expected, since structural Ca^2+^ is characterized by a higher affinity than catalytic [8]. 

We performed QM/MM simulations to characterize energetics of each of the six coordination interactions of the structural calcium cation (Appendix A). We found that the bond to His274 while being the weakest, still presents a free energy barrier of more than 6.5 kcal/mol to being replaced by a water molecule. What is more, this competing coordination state is a very shallow plateau lying 6.3 kcal/mol higher than the His274-coordinated state. This would mean that under physiological conditions such a rearrangement is highly unlikely and definitely should not be observed in equilibrium MD. The origins of such a discrepancy, if any, should be investigated in further work. It is not unexpected that a correct MM description of a structural Ca^2+^ in DFPase would require a bespoke set of parameters.

To date, only one other MD study has been performed on DFPase by the group of Professor Kamerlin in 2017 [11]. The authors used the OPLS-AA force field but did not observe issues presented herein. However, their setup was different: the protein was restrained in a layered scheme with the outermost layer of atoms almost completely frozen in place. What is more, all ionizable residues in the restrained region of the simulation were kept in their neutral forms, and the whole simulation was performed without periodic conditions and thus periodic electrostatics. Investigation into the possibility of influence of long-range electrostatics and dynamical rearrangements triggered by external parts of the enzyme on the structural Ca^2+^ site may be thus considered as starting points of further studies.

### 2.2. Origins of Structural Disturbances in the Catalytic Ca^2+^ Site

Catalytic Ca^2+^ site geometry was, in general, better described with all the residues involved mainly retaining their positions in space. However, even small discrepancies in the catalytic site architecture may lead to wrong results while studying substrate binding or reaction processes. The origins of such discrepancies may also hint at the existence of naturally occurring phenomena that were brought to light as a result of a skewed state equilibrium produced under a particular parameter set. That is why we studied the individual outcomes of MD trajectories in greater detail.

The combination of Amber19sb-DEF uniquely and wrongly leads to the involvement of backbone carbonyl oxygen atoms of N175 and D229 in the coordination of Ca^2+^ (Figure 2A). This state is also characterized by an additional coordinating water molecule absent in any X-ray structures. Changing to the DUM model significantly improves the quality of the modeling for all constituents of the active site but E21 (Figure 2B). Finally, the COM model produces results remarkably close to the X-ray coordinates (Figure 2C). HFE and IOD models can also produce a correct conformation. However, it happens with a lower probability with some of the replicas falling into erroneous conformations.

As was noted earlier, CHARMM36m combinations do not differ much in terms of Ca^2+^ site RMSD. Indeed, the exemplary resulting conformations are close to being indistinguishable (Figure 2D–F). They all share the switched conformation of E21 also observed for the Amber19sb-DUM system.

The default packaging of OPLS-AA/M produces two strikingly different dynamical outcomes. In the state *g* (Figure 1D) we can observe near-ideal sidechain coordinates correspondence with the reference for all residues but E21 which is again modeled in switched conformation. In contrast, state *h* presents a completely destroyed DFPase active site (Figure 1D and Figure 2G). Both active site asparagines lose their interactions with Ca^2+^ and their roles as coordinators are taken by a water molecule. E21 and D229 both over-coordinate Ca^2+^ in a forked conformation, hinting at a disproportionate contribution of electrostatics into metal coordination in OPLS-AA/M. S271 becomes the sixth coordinator; however, this involvement requires it to skew its backbone dramatically. Substituting DEF metals with DUM no longer produces this extreme state, however, and instead generates a new one (Figure 2I) that also deviates from the X-ray structure. In the state *i*, only N120 is excluded from the coordination sphere of Ca^2+^. We note that the same state is occasionally modeled by the Amber19sb-IOD combination.

From all the distorted conformations explored, only the one with switched E21 may be considered as marginally feasible. We thus decided to explore it in greater detail.

### 2.3. Origins of a Switched Conformation of E21

Shift between E21 sidechain oxygens as Ca^2+^ coordinators is a commonly recurring feature throughout various parameter sets tested (Figure 2B,D–F,H). We observed that in our simulations this conformation switch coincides with a formation of a hydrogen bond between the backbone carbonyl oxygen of A74 and nitrogen of G22 that is not detectable in any X-ray structure of DFPase. We used metadynamics to explicitly simulate this switch and evaluate its energetics. As a reference model, we used a QM/MM description of DFPase (Figure 3A). We found that the switched state of E21 indeed corresponds to a free energy minimum (Figure 3B). However, it is 4.5 kcal/mol higher than the global minimum corresponding to the crystallographic conformation and requires crossing of a barrier approximately 7 kcal/mol high. We found that only Amber19sb-COM can qualitatively reproduce this behavior while other parameter combinations result in apparent overstabilization of the G22-A74 hydrogen bond (Figure 3C–F and Appendix A). Our findings present an example of enzyme active site fine architectural features that rely on minor differences in interaction strength and therefore present a challenge for molecular modeling.

### 2.4. DFPase-DFP Binding/Unbinding Process

To this point, we established Amber19sb-COM as the most accurate parameter set across all that were tested. This metal model does not specifically limit the size of the MD integration step, and that makes this parameter combination a candidate for long production runs. To prove this, we performed a 1 μs simulation of the DFPase-DFP binding/unbinding process utilizing the funnel metadynamics technique. Our setup allowed us to observe multiple association/dissociation events (Figure 4A) which is crucial for the accuracy of binding free energy estimation. It should be noted that while the catalytic Ca^2+^ coordination shell was not artificially restrained, its geometry remained correct during the whole run.

The system was found to have two deep free energy minima corresponding to completely different binding modes of DFP (Figure 4B). The slightly deeper one corresponds to the pre-reaction state that was previously modeled only manually [11] (Figure 4C). In this work, we managed to obtain this state in an agnostic and physically correct way, thus providing additional support for the hypothesis of a general-base mechanism. This pose is close but not entirely identical to the one adopted by a dicyclopentylphosphoroamidate inhibitor in 2GVV structure (Appendix A) [5]. This is to be expected, however, since the inhibitor is bound in a reversed conformation with its amino group assuming the place and interactions of an attacking water molecule.

The second binding mode corresponds to DFP being held in a hydrophobic pocket on the side of the active site gorge (Figure 4D). In trajectory, this state generally precedes the transition into the pre-reaction one by substitution of a Ca^2+^-bound water molecule with phosphoryl oxygen. The water molecule then moves to the opposite direction of DFP, this way the whole binding process of DFP proceeds in what can be called a “counter-clockwise” way. This binding mode was not described earlier and no experimental evidence exists on its importance for the enzyme action. Probing how changing the properties of this hydrophobic binding site could affect the enzyme may be an interesting starting point for a future investigation. It is clear at this point that this finding should be taken into consideration in rational design endeavors including such of reversed enantioselectivity.

Two binding modes are separated by a high free energy barrier that corresponds to the replacement of a Ca^2+^-bound water molecule. It was noted before that such strongly bound water molecules present a major challenge in atomistic simulations of ligand binding by obstructing the sampling [30]. Despite the use of a collective variable to explicitly drive this process in the present study, it may still be suboptimal in a sense that it may not discriminate between different states with the same net amount of water molecules in a hydration shell. It may result in a simulation of a system transitioning between two minima via an unphysical route that therefore produces artificially heightened barriers. Another point of consideration was recently broad to attention regarding the lack of polarization and charge transfer in common MM force fields leading to the erroneous energetics even at the geometrically adequate transition state [31]. 

Unfortunately, no K_D_ was measured for the DFPase-DFP system and thus there is no proper reference value to judge the performance of our simulation. However, an upper limit for it is K_M_ and thus experimental binding free energy of DFP may be estimated to be not higher than −3.31 kcal/mol [13]. We obtained a value of −2.7 ± 0.2 kcal/mol which is an adequate agreement considering all the complications mentioned.

We demonstrate therefore that the simulation of substrate binding in DFPase is clearly a challenging task and may require sophisticated modeling setups to advance our understanding of it. QM/MM simulations of the transition between two free energy minima would allow us to reevaluate the energetics and reach a better agreement with the experiment. Since such a calculation may require a large QM subsystem, we look forward to the development of QM(ML)/MM hybrid schemes based on top-performing neural potentials [32,33,34]. Considering the currently available opportunities, our work leads to a strong recommendation of the Amber19sb force field supplemented with recent cation models by Li et al. as a modeling framework for DFPase.

While this work was mostly dedicated to the problem of modeling the substrate binding process, the chemical step also naturally draws attention of researchers. It can be studied by a variety of methods including those treating catalytic Ca^2+^ mechanically, namingly, EVB [35,36,37,38,39] and ReaxFF [40]. While it is common to derive broad ad hoc parameterizations in the ReaxFF framework, EVB mostly relies on default force field parameters. Consequently, the choice of metal cation parameters for EVB run could also influence the obtained results and as such pose a sizable problem. The extent of such influence, however, was not previously studied, and such work may be a valuable future addition to the overall EVB methodology.

## 3. Materials and Methods

All work was performed with Gromacs [41]. For MM simulations we used Gromacs 2021.1. For QM/MM simulations we used an interface between Gromacs and DFTB+ [42,43]. Both were patched with Plumed 2.7.1 [44].

### 3.1. Parameter Sets

Three freely available modern force fields were used: Amber19sb [45], CHARMM36m [46], and OPLS-AA/M [47,48]. We refer to pre-packaged parameterizations of Ca^2+^ ions within these force fields as DEF. It should be noted that in the case of CHARMM36m this parameterization includes NBfix paired potentials whereas DEF models for two other fields refer only to a particular 12-6 parameterization. 

Multisite models referred to as DUM were implemented as described in Duarte et al. [24]. COM, HFE, and IOD models refer to the parameterization strategy chosen while fitting their parameters [25]. The HFE model produces better results in reproducing hydration free energy, while IOD is tailored to produce more reliable coordination geometry. COM parameters balance these two objectives. While these parameters were derived for a number of three- and four-point water models, we use them in conjunction with tip3p-FB [49]. This is due to the demanding nature of substrate binding calculations that hardly allow for a dramatic increase in the number of simulated particles arising from using four- and five-point water models in bulk. Across three-point water models, tip3p-FB showed the best performance with these metal parameters.

A summary of metal parameters is given in Appendix A.

### 3.2. System Preparation

DFPase was modeled based on coordinates from PDB 3O4P [50]. Protonation states of residues were predicted with PROPKA and verified manually [51]. Histidine tautomer assignment and correction of amide flips were performed with Molprobity and also verified manually [52]. The system was placed in a cubic box with periodic boundary conditions and solvated with tip3p waters while modeling with DEF or DUM metals and with tip3p-FB otherwise. All crystallographic water molecules were retained while all added were manually filtered in case of incorrect solvation. Na^+^ and Cl^−^ ions were added to neutralize the net charge and reach 0.15M ionic strength. Systems were minimized with 5000 steps of steepest descent.

### 3.3. Molecular Dynamics

The equilibration phase consisted of seven steps. First, an NVT run of 100 ps was performed while positionally restraining heavy atoms by 1000 kJ/mol/nm^2^. A velocity rescale thermostat was utilized [53] for temperature coupling. Then, for five rounds of NPT equilibration of 100 ps, each restraint strength was gradually decreased as follows: 1000, 500, 200, 100, 10 kJ/mol/nm^2^ respectively. A stochastic barostat was used [54] for pressure control. Finally, a 50 ns unrestrained NPT run was performed. The last 10 ns were used to gather statistics on Ca2+ site conformations. For each parameter set, all equilibration steps were performed in 5 independent replicas (Appendix A). For all 12-6 metal models a 2 fs time step was used. For DUM, a 1 fs time step was used and the number of steps was scaled accordingly.

### 3.4. Analysis of Equilibration Dynamics

Catalytic Ca^2+^ site stability was assessed by calculating RMSD of the cation and 4 oxygen atoms in its coordination shell belonging to E21, N120, N175, and D229 respectively. For assessing structural Ca^2+^ site apart from the cation itself, three atoms were used: backbone carbonyl oxygen of L253, sidechain nitrogen of H274, and sidechain oxygen of D232 (Figure 1C). For both calculations, the system was aligned by the protein backbone.

### 3.5. QM/MM Simulations

For assessment of structural Ca^2+^ site stability (Appendix A) QM region consisted of 55 atoms and 3 link atoms. The sidechain of D232 and whole H274 with flanking backbone regions were included as well as two shells of water molecules. Well-tempered metadynamics was applied to the dRMSD collective variable constructed from six distances from the cation to its coordinators [55]. A potential 0.5 kJ/mol high was added once in 200 steps. The width was calculated adaptively according to a diffusion scheme on the basis of 100 previous steps. Biasfactor was set to 10. Individual free energy profiles along the distances to Ca^2+^ were constructed by performing Tiwary reweighting [56].

For the E21 conformation assessment, a relatively large QM system of 136 atoms and 13 link atoms was devised to minimize the force field bias from the MM part (Figure 3A). QM system size and composition is a common concern since it is the most probable cause of artifacts if designed incorrectly. It was shown for a metalloenzyme that qualitatively a system of 157 QM atoms is similar to one with 408 with MAD of 2.7 kcal/mol [57]. A somewhat opposing view is that at least QM/MM-calculated enzymatic reaction barriers are not highly sensitive to the size of the QM region as long as the first shell is included [58]. The later study is similar to one performed by us in the way that it relies on extensive QM sampling and not on single-point calculations. 

Metadynamics was performed along with two collective variables. The first one described the switch of the coordinating oxygen: d(E21_OE2_–Ca^2+^)–d(E21_OE1_–Ca^2+^). The second one described the substitution of the h-bonding partner of the G22 backbone nitrogen: d(G22_N_–E21_OE1_)–d(G22_N_–A74_O_). Potential height, width, and pace as well as the bias factor were identical to the ones described in the previous paragraph.

In both setups, the QM part was described with the DFTB3 method with 3ob-3-1 parameter set [59,60]. DFTB3 was previously shown to produce good geometries and adequate energy for a wide range of scenarios. It performed well for biomolecular systems [61,62,63], systems with metal cations in general [29,64,65,66], and Ca^2+^ specifically [67,68], achieving better energies then DFT calculations with a double-ζ basis set. In both tasks, QM simulation was run in parallel by 10 walkers each sampling for 100 ps thus accumulating 1 ns of QM/MM trajectory. 0.5 fs time step was used.

QM/MM calculations were carried out using the equipment of the shared research facilities of HPC computing resources at the Lomonosov Moscow State University [69].

### 3.6. MM Simulations of E21 Conformation

Loss of E21-G22 h-bond was pronounced even in the course of restrained NPT runs. Thus, we speculated that this phenomenon is not a consequence of a disruption of the structural Ca^2+^ site that was prominent under every simulation setup and therefore can be studied individually. To achieve that we performed our metadynamics simulations while positionally restraining the backbone by applying 100 kJ/mol/nm^2^ potential. Simulations were run for 50 ns with a 1 fs time step for the DUM metal model and 2 fs otherwise.

Metadynamics setup was similar to the one used for the QM/MM simulation of E21. The pace of potential addition was set to 2000 steps, and adaptive potential width was calculated from 1000 steps.

### 3.7. Funnel Metadynamics

The funnel metadynamics setup was similar to the one we previously used [61]. Protein was kept in place by positionally restraining backbones of beta strands second to the most external ones in each of the blades by 100 kJ/mol/nm^2^ potential. Structural Ca^2+^ was also positionally restrained to prevent artifacts arising from its poor description. Metadynamics potential of 1 kJ/mol was added once in 2000 steps with width calculated adaptively from 1000 steps using a diffusion scheme. Two collective variables were used. The first one was the distance between the phosphoryl oxygen of DFP and catalytic Ca^2+^. The second was the number of water molecules surrounding DFP as calculated by using the 12-6 switch function with r_0_ set to 0.3 nm. The use of such a variable was envisioned to speed up water replacement in the binding site as the ligand moves into it. 

The use of funnel metadynamics setup requires the correction of the resulting energy differences to turn it into a binding free energy estimate. The correction originates from the nature of the funnel-shaped restraint. Effective sampling is achieved by confining the unbound ligand within only a narrow cylinder thus restraining it from diffusing through all the solvent phase. While this dramatically reduces the simulation time needed to achieve recrossing, it comes at a price of unaccounted entropic contribution to the energy of an unbound state that naturally depends on the radius of the cylinder. The equation to calculate a correction to account for this contribution can be found in [15]. In the present study for a cylinder of a 1 Å radius the correction amounts to 3.8 kcal/mol.

We report the final calculated binding free energy together with an error estimation similar to one used in [70]. The width of the window for statistical analysis was 100 ns.

DFP parameters were produced with the parameterize tool by tailoring GAFF2 torsions towards QM scans on the wB97X-D level of theory with aug-cc-pVDZ basis set [71]. Charges were assigned with the RESP protocol by considering five different conformations of DFP.

## 4. Conclusions

We have compared the ability of 15 MM parameter sets to correctly model calcium-binding sites in DFPase. These sites proved to be particularly challenging systems to describe by means of classical molecular modeling. This was found to be especially true for structural Ca^2+^, and we presented a clear need for further research in this direction. Of all parameter sets tested the combination of Amber19sb with recently developed 12-6 Ca^2+^ models produces the best results and therefore is recommended for any further molecular modeling studies of DFPase. We used it to model the DFP binding process for the first time and described a previously unreported binding mode in a hydrophobic side pocket of DFPase. Our work lays the methodological foundation for long simulations of this enzyme to produce new insights and inform rational design strategies.

## Figures and Tables

**Figure 1 molecules-26-05839-f001:**
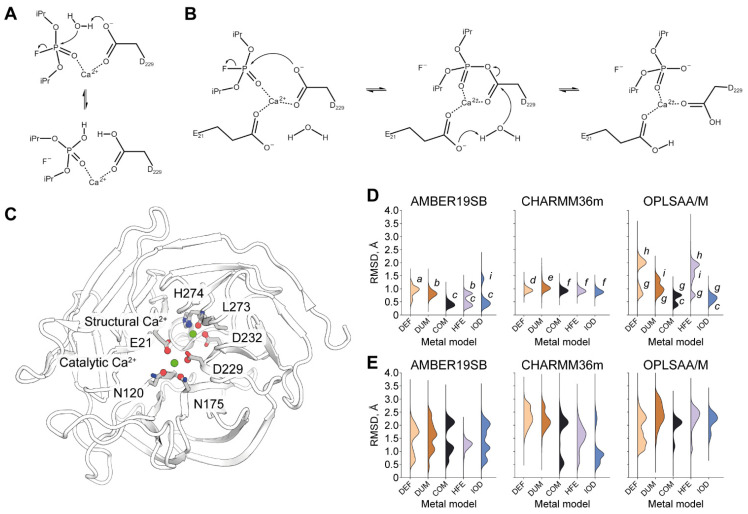
Overview of DFPase reaction, calcium-binding sites and modeling of their dynamic behavior. (**A**). A scheme of the proposed general-base reaction mechanism. (**B**) A scheme of the proposed nucleophilic reaction mechanism. (**C**) DFPase structure. Coordinating residues are highlighted. Atoms involved in RMSD calculation are shown as spheres. (**D**). Stability of catalytic Ca^2+^ site. Individual states of interest are marked as lowercase letters to be referenced later on in text and figures. (**E**). Stability of structural Ca^2+^ site. Each violin plot represents RMSD over the last 10 ns of 5 replicas for each parameter combination. Explanation of system naming can be found in the Materials and Methods section.

**Figure 2 molecules-26-05839-f002:**
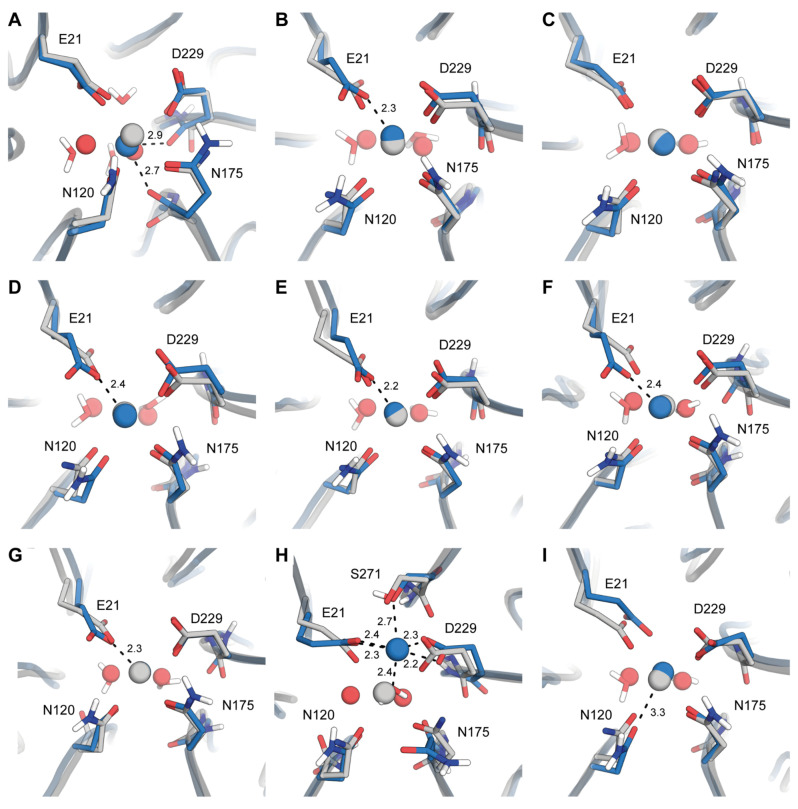
Structural disturbances in catalytic Ca^2+^ site of DFPase when modeled with different parameter combinations. (**A**) State *a* showcased on Amber19sb-DEF results. (**B**) State *b*, Amber19sb-DUM. (**C**) State *c*, Amber19sb-COM. (**D**) State *d*, CHARMM36m-DEF. (**E**) State *e*, CHARMM36m-DUM. (**F**) State *f*, CHARMM36m-COM. (**G**) State *g*, OPLS-AA/M-DEF. (**H**) State *h*, OPLS-AA/M-DEF. (**I**) State *i*, OPLS-AA/M-DUM. Modeled systems are shown in blue, shown in gray are coordinates from PDB ID 3O4P. Non-polar hydrogen atoms and outer dummy atoms in DUM models are omitted for clarity.

**Figure 3 molecules-26-05839-f003:**
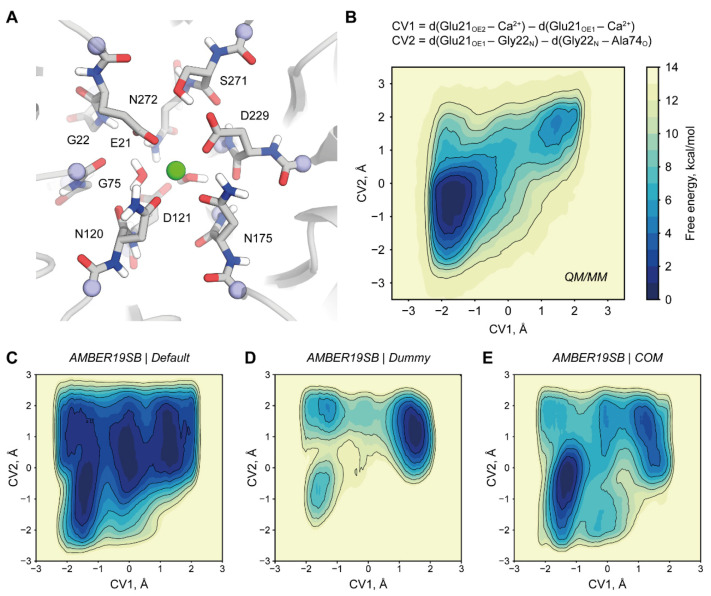
Competition for a hydrogen bond with G22 and its influence on the E21 conformation studied with QM/MM and MM treatment. (**A**) Composition of QM subsystem. Linking atoms are shown in blue. (**B**) QM/MM free energy profile of E21 conformation switch. (**C**–**E**) Free energy profiles of Amber19sb systems showing the benefit of using the COM metal model to approach QM/MM reference. Similar data for other force fields are shown in Appendix A. Non-polar hydrogen atoms are omitted for clarity.

**Figure 4 molecules-26-05839-f004:**
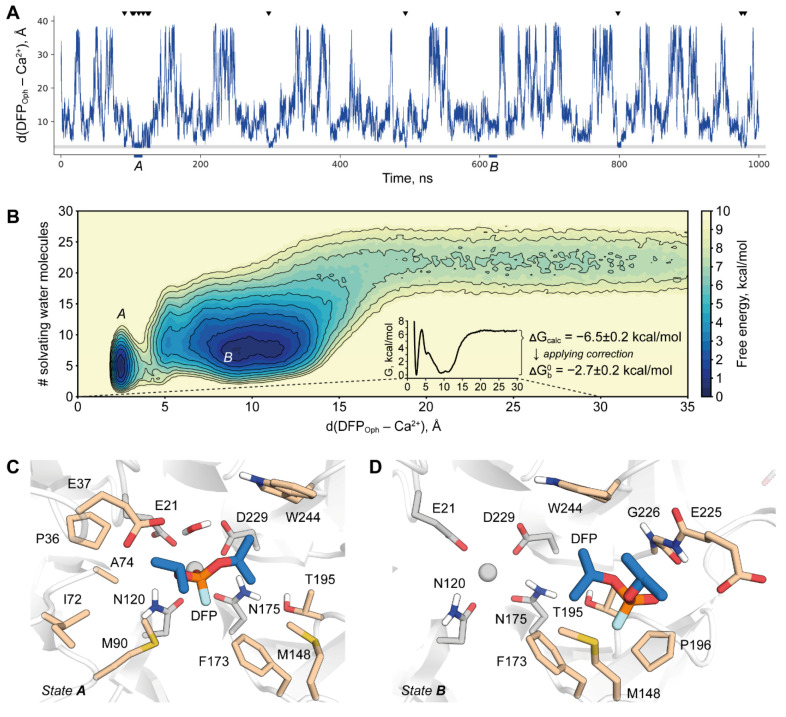
Funnel metadynamics simulation of DFPase-DFP interaction upon binding. (**A**) Time progression of the simulation. Shaded gray area corresponds to Ca^2+^ coordination by phosphoryl oxygen. Black arrows indicate reaching the pre-reaction state. Time ranges used to extract frames of binding modes of interest are marked with capital letters. (**B**) Free energy profile of substrate binding. Minima corresponding to stable binding modes are marked with capital letters. (**C**) Pre-reaction state reached in the simulation. (**D**) The alternative stable state reached in the simulation. DFP carbon atoms are shown in blue, catalytic residues are colored gray and hydrophobic residues forming binding pockets for both states are shown in wheat. Non-polar hydrogen atoms are omitted for clarity.

## Data Availability

Plumed input files and force field parameters are available online at https://vsb.fbb.msu.ru/share/2021/dfpase-Ca.

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
