# Peer review of "Probing the Suitability of Different Ca2+ Parameters for Long Simulations of Diisopropyl Fluorophosphatase"

_molecules, 2021, doi:10.3390/molecules26195839_

Round 1

Reviewer 1 Report

The present manuscript is devoted to the validation of different sets of parameters for the study of the behavior of calcium ions in a diisopropyl fluorophosphatase. The authors use different classical and hybrid QM/MM methodologies to investigate into the dynamics of the structural and catalytic ions. In general the work has been well done and the results are of interest for the simulation and investigation of this and similar systems. There are however, a number of issues that should be addressed by the authors in a revised version:

  • The authors mention ‘x-ray structures’ in plural, but the comparison with computational simulations is always done taking as reference the 3O4P structure. In order to put into perspective the variations observed during the MD simulations it is necessary to discuss first the variations observed (if any) also in the x-ray structures as a function of the ligand employed or any other change introduced during the crystallization process.
  • The authors also use as a reference for the classical simulations the results of QM/MM molecular dynamics. While the performance of the QM method is briefly discussed, the authors should discuss in more detail the reliability of these QM/MM calculations and the possible artifacts.
  • The authors observe two binding modes separated by a high energy barrier. It is quite clear that the observed free energy profile seems a consequence of a incomplete choice of the reaction coordinate. However, this does not produce heightened barriers. In fact, an incomplete choice of the reaction barrier usually results in a too lower free energy barrier because the work to be done on that coordinate is not considered.
  • Regarding the estimation of the binding free energy, the authors apply a correction that amounts to a large fraction of the calculated magnitude. This correction must be more clearly explained. Also, about this value, the use of two decimal figures seems a too optimistic view of the statistical error. The authors should try to provide an estimation of this.
  • ‘linking atoms’ must be changed to ‘link atoms’

Author Response

We express our gratitude to the reviewer for his attention to the manuscript and valuable remarks. Below is the point-by-point response to the commentaries received.

Point 1:

The authors mention ‘x-ray structures’ in plural, but the comparison with computational simulations is always done taking as reference the 3O4P structure. In order to put into perspective the variations observed during the MD simulations it is necessary to discuss first the variations observed (if any) also in the x-ray structures as a function of the ligand employed or any other change introduced during the crystallization process.

Response

We added a graphic and numerical comparison of all 7 WT DFPase structures available to date, as Figure S1 and Table S1 and S2 respectively. We also added a visual comparison of the binding mode of DFP obtained by our funnel metadynamics and a binding mode of an organophosphorus inhibitor from structure 2GVV as Figure S6.

Point 2:

The authors also use as a reference for the classical simulations the results of QM/MM molecular dynamics. While the performance of the QM method is briefly discussed, the authors should discuss in more detail the reliability of these QM/MM calculations and the possible artifacts.

Response

The performance of the QM method used was extensively studied in references 61, 62, 63 (biomolecular systems), 29, 64, 65, 66 (metal ions), 67, 68 (calcium cation specifically). We expanded the section of the materials and methods with these references to add clarity. The most possible artifacts in such a study could arise from the size and composition of the QM region, and this is the reason why it was designed to be relatively large. It was shown for a metalloenzyme that qualitatively a system of 157 QM atoms is similar to one with 408 with MAD of 2.7 kcal/mol (Liao and Thiel, 2013, JCC). A somewhat opposing view is that at least QM/MM-calculated enzymatic reaction barriers are not highly sensitive to the size of the QM region as long as the first shell is included (Jindal and Warshel, 2016, JPCB). The later study is similar to one performed by us in a way that it relies on extensive QM sampling and not on single-point calculations. We added this paragraph into the manuscript.

Point 3:

The authors observe two binding modes separated by a high energy barrier. It is quite clear that the observed free energy profile seems a consequence of a incomplete choice of the reaction coordinate. However, this does not produce heightened barriers. In fact, an incomplete choice of the reaction barrier usually results in a too lower free energy barrier because the work to be done on that coordinate is not considered.

Response:

The point is valid, and we agree that in most cases where the transition is successfully modeled the barrier will indeed be lowered due to the projection issues (as in Bussi and Laio, 2020, Nat. Rew. Phys., Fig2B,E). In these cases, however, the TS configuration sampled is correct (meaning close to what occurs in reality). In our case what we wanted to express was the concern that our simulation setup could, with a marginal probability, lead to the transition occurring through the wrong TS associated with such a high barrier. This, in turn, could be a consequence of (more probable) limitations of classic MD or (less but still probable) redundancy across the second driving collective variable (# of waters). We rephrased this place in the manuscript to make it more clear what was the concern exactly.

Point 4:

Regarding the estimation of the binding free energy, the authors apply a correction that amounts to a large fraction of the calculated magnitude. This correction must be more clearly explained. Also, about this value, the use of two decimal figures seems a too optimistic view of the statistical error. The authors should try to provide an estimation of this.

Response:

The correction originates from the nature of the funnel-shaped restraint that is the sole idea behind funnel metadynamics (Limongelli et. al, 2013, PNAS). Effective sampling is achieved by confining the unbound ligand within only a narrow cylinder thus restraining it from diffusing through all the solvent phase. This trick dramatically reduces the simulation time needed to achieve recrossing, that is, a full unbound-bound-unbound cycle. This comes at a price of unaccounted entropic contribution to the energy of an unbound state that naturally depends on the radius of the cylinder. The equation to calculate a correction to account for this contribution can be found in (Limongelli et. al, 2013, PNAS), and in the case of a cylinder of a 1A radius is 3.8 kcal/mol. We added this paragraph to the Materials and Methods section of the manuscript.

We also performed an error estimation similar to (Bhakat and Söderhjelm, 2017, J Comput Aided Mol Des) and updated values reported in the text.

Point 5:

‘linking atoms’ must be changed to ‘link atoms’

Response: 

We changed all occurrences of this phrase in the text.

Reviewer 2 Report

Referee report concerning the manuscript

molecules-1376275

Probing the suitability of different Ca2+ parameters for long simulations of diisopropyl fluorophosphatase

by A. Zlobin et al.

The authors studied the effect of various force fields on the geometric parameters of the Michaelis complex of the enzyme diisopropyl fluorophosphatase with the substrate. Special attention was given to the parametrization of two calcium ions that are present at the active site. All together 15 different force field combinations were considered. Based on geometric and energetic parameters the authors suggested that AMBER19sb force field along with the 12-6 parametrization for calcium yields the most promising results. Presented work is important for parametrization in future studies of catalytic step of this enzyme. The manuscript represent an important new contribution and it should be published after a minor revision.   There are however few weak points that the authors may consider.

1) I would add a scheme for chemical step of this enzyme and give an overview of experimental kinetics.

2)  A very strict test for the calcium ion parametrization is calculation of free energy of hydration. The experimental value is -312 kcal/mol (Y. Marcus, J. Chem. Soc. Farad. Trans. 1991,87, 2995)

Look also the thermodynamic integration study of Ca++ hydration by Straatsma and Berendsen, JCP 89, 1988,5876.

A challenge for future is to calculate this value for various Ca++ parametrization in conjunction with various water models. I would definitively mention the effects of long term electrostatics, Born correction in the case of spherical cutoff etc.

Quote, comment!

3) page 2

All atom and united atom force field MD cannot be integrated with a time step longer than 2fs, while you wrote 4 fs. Check, comment!

4) I would add a table/figure with Ca++ nonbonding parameters and functional forms of the force field applied in this study.

5) A real challenge for future is to address the chemical step by various sets of nonbonding parameters. Software is developed and ready to be used free of charge.

Take a look:

Bauer, P., Barrozo, A., Purg, M., Amrein, B.A., Eguerra, M., Barrie Wilson, P., Major, D.T., Åqvist, J., Kamerlin, S.C.L. (2018) SoftwareX DOI: 10.1016/j.softx.2017.12.001 "Q6: Acomprehensive toolkit for empirical valence bond and related free energy calculations".

https://www.icm.uu.se/cbbi/aqvist-lab/q/

Quote, comment!

6) EVB is the method of choice and catalytic effect comes out automatically when one compares the activation energy in the enzyme with the corresponding reaction in water. I have an impression that the catalytic effect will be pretty much the same with all different force fields since substantial cancellation of errors is anticipated.

Warshel, Computer Modelling of Chemical Reactions in Enzymes

and Solutions, John Wiley and Sons, 1991, New York

Proc Natl Acad Sci U S A. 2019 Jan 8;116(2):389-394

PROTEINS: Structure, Function, and Bioinformatics, 82 (2014) 3347-3355.

Quote, comment!

7) One more thing:  in computational enzymology one always applies substantial values of harmonic position restraints in order to avoid gross conformational changes. Look for computational details in the reference 11 (Purg et al). In this respect the geometrical parameters reported in this study are less relevant. See also previous point.

Comment!

--End of comments--

Author Response

We express our gratitude to the reviewer for his attention to the manuscript and valuable remarks.

Below is the point-by-point response to the commentaries received.

Point 1:

 I would add a scheme for chemical step of this enzyme and give an overview of experimental kinetics.

Response: 

A scheme of two rivaling mechanisms was added to Figure 1.

Point 2:

A very strict test for the calcium ion parametrization is calculation of free energy of hydration. The experimental value is -312 kcal/mol (Y. Marcus, J. Chem. Soc. Farad. Trans. 1991,87, 2995)

Look also the thermodynamic integration study of Ca++ hydration by Straatsma and Berendsen, JCP 89, 1988,5876.

A challenge for future is to calculate this value for various Ca++ parametrization in conjunction with various water models. I would definitively mention the effects of long term electrostatics, Born correction in the case of spherical cutoff etc.

Response:

This type of validation for all the ion parameters used was performed by the authors of these parameters in ref 24, 25, 45-48. We provide full references to these papers in the text. We do not make claims of universal applicability, nor did we parameterize cation parameters ourselves in the study and thus fully rely on the information provided by the authors of parameterizations in ref 24, 25, 45-48.

Point 3:

page 2, All atom and united atom force field MD cannot be integrated with a time step longer than 2fs, while you wrote 4 fs. Check, comment!

Response: 

Some schemes relying on constraints, mass repartitioning or turning hydrogens into dummies provide a technical possibility for time steps as long as 7fs (Feenstra et. al, 1999, JCC; Jung and Siguta, 2020, JCP). While this indeed is not the most widespread choice, it may be utilized (as a matter of fact, it is a default in the AceMD engine. Harvey et. al, 2009, JCTC). We added these references to the manuscript and rephrased the context to specifically mention that timesteps higher than 2fs are special schemes that may or may not be considered by an MD practitioner.

Point 4:

I would add a table/figure with Ca++ nonbonding parameters and functional forms of the force field applied in this study.

Response:

 We added a table with parameters in the supplementary information section.

Point 5 and 6:

5) A real challenge for future is to address the chemical step by various sets of nonbonding parameters. Software is developed and ready to be used free of charge.

Take a look: Bauer, P., Barrozo, A., Purg, M., Amrein, B.A., Eguerra, M., Barrie Wilson, P., Major, D.T., Åqvist, J., Kamerlin, S.C.L. (2018) SoftwareX DOI: 10.1016/j.softx.2017.12.001 "Q6: Acomprehensive toolkit for empirical valence bond and related free energy calculations".

https://www.icm.uu.se/cbbi/aqvist-lab/q/

6) EVB is the method of choice and catalytic effect comes out automatically when one compares the activation energy in the enzyme with the corresponding reaction in water. I have an impression that the catalytic effect will be pretty much the same with all different force fields since substantial cancellation of errors is anticipated.

Warshel, Computer Modelling of Chemical Reactions in Enzymes

and Solutions, John Wiley and Sons, 1991, New York

Proc Natl Acad Sci U S A. 2019 Jan 8;116(2):389-394

PROTEINS: Structure, Function, and Bioinformatics, 82 (2014) 3347-3355.

Response: 

The chemical step naturally draws attention and can be studied by a variety of methods. For those treating catalytic Ca2+ mechanically (EVB, ReaxFF) the choice of parameters could indeed influence the obtained results. The extent of such influence however was not previously studied, and such work may be a valuable future addition to the overall EVB methodology.

We added this notion into the text as well as references to the methods.

Point 7:  

One more thing:  in computational enzymology one always applies substantial values of harmonic position restraints in order to avoid gross conformational changes. Look for computational details in the reference 11 (Purg et al). In this respect the geometrical parameters reported in this study are less relevant. See also previous point.

Comment!

Response:

The main objective of this study, as we also state at the beginning of the reply, is to come closer to the ability to correctly simulate the binding stage. Imposing arbitrary restraints to force an ill-behaving force field to correctly simulate some states for which we have a reference (e.g. apo form of enzyme model from Xray study) may result in it incorrectly describing some other states or transitions between them. While this is also true for choosing the force field judging by the correctness of describing these very states, we rely on the fact that force field parameterization is performed holistically, without respect for this particular system and our knowledge about it, and therefore this provides more strong justification. In a sense, it is a glimpse of the overfitting problem.

When simulating the chemical step this indeed may not be relevant since a researcher is not interested in the complete space of configurations the system may adopt, even given that the simulation framework is reliable enough for us to believe the ensemble it produces. In the case of EVB one is interested in probabilities of somewhat predetermined steps along the path that was arbitrarily set. What is more, this happens in only the subset of protein states that is also arbitrarily specified - indeed, there is no need to allow product or reagent to leave the site, or protein unfold. In studying the landscape of bound states it is, however, not so, since the space of hypotheses that we should consider is higher. As an example, we should not rule out the possibility of induced fit and thus should not positionally restrain the exterior of the protein even if it adopts the same conformation in each X-ray structure. That said, one of the directions that can also be undertaken from this study is to address the problem of the structural Ca2+ since the correctness of its restraining is the very same arbitrary hypothesis and not a fact. We have expressed this concern in the conclusions of the original manuscript.